# *Neospora caninum*: Differential Proteome of Multinucleated Complexes Induced by the Bumped Kinase Inhibitor BKI-1294

**DOI:** 10.3390/microorganisms8060801

**Published:** 2020-05-26

**Authors:** Pablo Winzer, Joachim Müller, Dennis Imhof, Dominic Ritler, Anne-Christine Uldry, Sophie Braga-Lagache, Manfred Heller, Kayode K. Ojo, Wesley C. Van Voorhis, Luis-Miguel Ortega-Mora, Andrew Hemphill

**Affiliations:** 1Institute of Parasitology, Department of Infectious Diseases and Pathobiology, Vetsuisse Faculty, University of Bern, Länggass-Strasse 122, 3012 Bern, Switzerland; pablo.winzer@vetsuisse.unibe.ch (P.W.); dennis.imhof@vetsuisse.unibe.ch (D.I.); dominic.ritler@vetsuisse.unibe.ch (D.R.); 2Graduate School for Cellular and Biomedical Sciences, University of Bern, Mittelstrasse 43, 3012 Bern, Switzerland; 3Proteomics & Mass Spectrometry Core Facility, Department for BioMedical Research (DBMR), University of Berne, Freiburgstrasse 15, CH-3010 Berne, Switzerland; anne-christine.uldry@dbmr.unibe.ch (A.-C.U.); sophie.lagache@dbmr.unibe.ch (S.B.-L.); manfred.heller@dbmr.unibe.ch (M.H.); 4Center for Emerging and Re-emerging Infectious Diseases (CERID), Division of Allergy and Infectious Diseases, Department of Medicine, University of Washington, Seattle, WA 98109, USA; ojo67kk@yahoo.ca (K.K.O.); wesley@uw.edu (W.C.V.V.); 5SALUVET, Animal Health Department, Faculty of Veterinary Sciences, Complutense University of Madrid, Ciudad Universitaria s/n, 28040 Madrid, Spain; luisucm@ucm.es

**Keywords:** antigenic variation, chemotherapy, drug adaptation, drug resistance

## Abstract

Background: the apicomplexan parasite *Neospora caninum* causes important reproductive problems in farm animals, most notably in cattle. After infection via oocysts or tissue cysts, rapidly dividing tachyzoites infect various tissues and organs, and in immunocompetent hosts, they differentiate into slowly dividing bradyzoites, which form tissue cysts and constitute a resting stage persisting within infected tissues. Bumped kinase inhibitors (BKIs) of calcium dependent protein kinase 1 are promising drug candidates for the treatment of *Neospora* infections. BKI-1294 exposure of cell cultures infected with *N. caninum* tachyzoites results in the formation of massive multinucleated complexes (MNCs) containing numerous newly formed zoites, which remain viable for extended periods of time under drug pressure in vitro. MNC and tachyzoites exhibit considerable antigenic and structural differences. Methods: Using shotgun mass spectrometry, we compared the proteomes of tachyzoites to BKI-1294 induced MNCs, and analyzed the mRNA expression levels of selected genes in both stages. Results: More than half of the identified proteins are downregulated in MNCs as compared to tachyzoites. Only 12 proteins are upregulated, the majority of them containing SAG1 related sequence (SRS) domains, and some also known to be expressed in bradyzoites Conclusions: MNCs exhibit a proteome different from tachyzoites, share some bradyzoite-like features, but may constitute a third stage, which remains viable and ensures survival under adverse conditions such as drug pressure. We propose the term “baryzoites” for this stage (from Greek βαρυσ = massive, bulky, heavy, inert).

## 1. Introduction

The phylum Apicomplexa parasites includes important pathogens for humans and animals. Among those, *Neospora caninum*, which is closely related to *Toxoplasma gondii*, causes abortion, stillbirth or birth of weak offspring in cattle, sheep and other ruminants, and also infects a wide range of other species [1]. Canines act as definitive hosts of *N. caninum*, and sexual development of the parasite in the intestinal tissue leads to the formation of oocysts, which are shed with the feces. Canines can also act as intermediate hosts and are affected by neurological symptoms [2]. Three stages are important for the life cycle of *N. caninum,* sporozoites, tachyzoites, and bradyzoites. Sporozoites develop within the oocysts and are orally infective. Tachyzoites represent the rapidly proliferating disease-causing stage, which infects numerous cell types and tissues, and can also cross the placenta and affect the developing fetus, which leads to malformations and/or abortion. The slowly proliferating bradyzoites are formed upon the immunological and physiological host response, and they represent a tissue cyst-forming stage that persists within infected tissues for extended periods. A temporal immunomodulation, such as during pregnancy, frequently leads to recrudescence and re-differentiation into tachyzoites which infect the developing fetus [3]. No vaccine is currently licensed for the prevention of bovine or canine neosporosis, and specific immuno- or chemotherapeutical treatments are lacking [4,5]. As a consequence, there is a keen interest in novel chemotherapeutics acting on specific targets [6].

Examples for such targets are calcium-dependent protein kinases (CDPKs). The genes coding for these kinases originate from plants [7], and therefore constitute interesting drug targets. During the last decade, CDPK1 has been an important focus as a target for drug development against a wide range of apicomplexans such as *Plasmodium falciparum*, *Toxoplasma gondii*, *Neospora caninum*, *Sarcocystis neurona*, *Besnoitia besnoiti*, *Babesia bovis*, *Theileria equi* and *Cryptosporidium parvum* [7,8]. Bumped kinase inhibitors (BKIs) are ATP-competitive kinase inhibitors that target CDPK1 in different apicomplexans [9]. They exhibit a high degree of efficacy and specificity for apicomplexan CDPK1 and are optimized to fit into a hydrophobic pocket within the ATP binding site. BKIs inhibit the ATP binding activity of a range of CDPK1 isoforms, including those of *N. caninum* [10]. They were shown to block invasion of invasive stages into host cells [10] and also inhibit microneme secretion, and thus invasion and egress [11]. Several BKIs have been studied so far with respect to efficacy against *N. caninum* infection, among them the pyrazolopyrimidine BKI-1294. This compound has shown highly promising efficacy in pregnant mouse models for neosporosis [12] and for toxoplasmosis [13], and in a pregnant sheep model for toxoplasmosis [14]. Moreover, other BKIs are effective against neosporosis, as shown in pregnant mouse [15] and sheep [16] models.

BKI-1294 does not act parasiticidal in vitro [17]. Prolonged exposure of mammalian cells infected with different *Neospora* isolates (Nc-1, Nc-Liv, and Nc-Spain7) and with the *T. gondii* strains RH and ME49 results in the formation of intracellular multinucleated complexes (MNCs) [12,17]. In addition, we found that BKI-1294-induced MNCs exhibited a deregulated gene expression pattern, as evidenced by the simultaneous expression of bradyzoite and tachyzoite antigens [12]. Immunofluorescence staining showed that the formation of MNCs during extended BKI-1294 treatment was accompanied by a lack of SAG1 surface expression and a decreased CDPK1 expression, and the formation of numerous zoites incapable of disjoining from each other. Following drug removal, proliferation continued, and zoites lacking NcSAG1 emerged from the periphery of these complexes, forming infective tachyzoites after 10 days [18]. These results suggested that MNCs could constitute a drug-induced resting stage that assures the survival of the parasite until the drug pressure is released.

To address this question, we have chosen proteomics as an appropriate tool. A number of studies had been carried out on the proteome of *N. caninum* previously. As one of the first methods, 2-D-Differential Gel electrophoresis (DIGE) was applied to define proteomic changes during tachyzoite-to-bradyzoite differentiation, identifying 20 proteins upregulated in tachyzoites and 6 in bradyzoites. However, this approach had technical limitations in that membrane proteins were not very well resolved [19]. Furthermore, proteomics was applied to identify potential virulence factors as exemplified by a study compared two isolates, namely Nc-Spain1H (low virulence) and Nc-Spain7 (high virulence) during different steps of the lytic cycle [20]. To investigate BKI-1294-induced MNCs in more detail, we thus compared the proteomes of isolated tachyzoites to the proteomes of MNCs by shotgun mass spectrometry.

## 2. Materials and Methods 

### 2.1. Tissue Culture Media, Biochemicals, and Drugs

If not stated otherwise, all tissue culture media were purchased from Gibco-BRL (Zürich, Switzerland), and biochemicals from Sigma (St. Louis, MO, USA). BKI-1294 was synthesized to >99% purity by HPLC and NMR by Wuxi AppTech, Inc. (Wuhan, China) and provided by the Center for Emerging and Reemerging Infectious Diseases (CERID), Division of Allergy and Infectious Diseases, Department of Medicine, University of Washington (Seattle, WA, USA), and was stored in powder form, dry and protected from light at room temperature. For in vitro studies, 20 mM stock solutions were prepared in dimethyl sulfoxide (DMSO) and they were stored at −20 °C. Primers for real-time PCR (RT-PCR) were purchased from Eurofins (Luxemburg, Luxemburg).

### 2.2. Host Cell Cultivation and Parasite Maintenance

Human foreskin fibroblasts (HFF; ATCC^®^ SCRC-1041™) were maintained in Dulbecco’s modified Eagle medium (DMEM), containing phenol red, supplemented with 10% heat-inactivated and fetal calf serum (FCS), 100 U of penicillin/mL, and 100 µg streptomycin/mL. The *Neospora caninum*-Spain7 (Nc-Spain7) isolate was maintained by infecting semi-confluent HFF monolayers and cultivated at 37 °C and 5% CO_2_, with passages once or twice per week as previously described [21]. For induction of the formation of MNCs, *N. caninum* infected HFF were treated with 5 µM BKI-1294 during 4–6 days.

### 2.3. Microscopy

Transmission electron microscopy (TEM) of BKI-1294-treated or non-treated Nc-Spain7-infected HFF was done as previously described [12,17]. In short, infected monolayers grown in T-25 tissue culture flasks were fixed in 2.5% glutaraldehyde in 100 mM sodium cacodylate buffer pH 7.3, removed from the flask by scraping, post-fixed in 2% osmium tetroxide in cacodylate buffer, and sequentially dehydrated in a graded (30-50-70-90-100%) ethanol series. Specimens were embedded in Epon 812 epoxy resin, polymerized at 65 °C, and ultrathin sections were cut on a Reichert and Jung ultramicrotome (Reichert & Jung, Vienna). Following transfer onto 300 mesh formvar-carbon coated nickel grids (Plano GmbH, Marburg, Germany), they were stained in Uranyless™ and lead citrate (EMS, Hatfield, PA, USA), and specimens were inspected on a CM12 TEM operating at 80 kV. 

For scanning electron microscopy (SEM), semi-confluent HFF monolayers were maintained in T175 cell culture flasks and were infected with Nc-Spain7 tachyzoites. At 4 h post infection, treatments with 5 µM BKI-1294, or the corresponding amount of DMSO in controls were initiated, and cultures were maintained at 37 °C/5% CO_2_ during 3 (controls) or 6 days (BKI-1294-treated cultures), until MNCs were formed. Subsequently, the infected monolayers were washed twice with PBS, followed by removal of infected cells with a rubber cell scraper and resuspension in PBS. After passaging the suspensions twice through a 25-gauge needle to break the host cells, MNCs and tachyzoites were separated from host cell debris by Sephadex G-25 chromatography. The parasite fractions in the flow through were collected by centrifugation (15 min, 1000× *g*, 4 °C), washed twice with PBS, and were fixed in 2.5% glutaraldehyde and 2% osmium tetroxide, and dehydrated in ethanol as for TEM. They were finally washed twice in hexamethyl-disilazane (HDS), taken up in a small volume of HDS, and were allowed to settle onto glass coverslips under a fume hood. They were then sputter-coated with gold and inspected on a JEOL 840 SEM operating at 25 kV. 

### 2.4. Proteomics

Semi-confluent HFF monolayers were maintained in T175 cell culture flasks, and were infected with Nc-Spain7 tachyzoites. At 4 h post infection, treatment with 5 µM BKI-1294, or the corresponding amount of DMSO in controls, was initiated, and cultures were maintained at 37 °C/5% CO_2_ during 3 (controls) or 6 days (BKI-1294-treated cultures), until multinucleated complexes (MNCs) were formed. Subsequently, the infected monolayers were washed twice with PBS, followed by removal of infected cells with a rubber cell scraper and resuspension in PBS. After passaging the suspensions twice through a 25 gauge needle to break the host cells, MNCs and tachyzoites were separated from host cell debris by Sephadex G-25 chromatography. The parasite fractions in the flow through were collected by centrifugation (15 min, 1000× *g*, 4 °C) and washed twice with PBS, as described [21].

Cell pellets were lysed in 100 μL 8M urea/100 mM Tris/HCl pH 8/cOmplete^TM^ protease inhibitor cocktail (Roche Diagnostics, Rotkreuz, Switzerland) by incubation for 15 min at RT followed by 15 min in an ultrasonic water bath. Proteins were reduced and alkylated with 10 mM DTT for 30 min at 37 °C and 50 mM iodoacetamide for 30 min at 37 °C. Proteins were precipitated at −20 °C by addition of 5 volumes cold acetone and incubation at −20 °C overnight. All liquid was carefully removed, and the pellet dried in ambient air for 15 min before reconstitution of proteins in 200 μL of 8 M urea, 50 mM Tris-HCl pH 8.0. Protein concentration was determined by Bradford assay and an aliquot corresponding to 10 μg protein was digested by trypsin (1:50 trypsin/protein ratio) for 6 hours at 37 °C after dilution of urea concentration to 1.6 M with 20 mM Tris-HCl pH 8.0 and 2 mM CaCl_2_. The digests were acidified with TFA (1%) and analyzed by LC-MS/MS. Three repetitive injections of an aliquot corresponding to 500 ng of protein digest was separated on an EASY-nLC 1000 coupled to a QExactive mass spectrometer (ThermoFisher Scientific). Peptides were trapped on an Acclaim PepMap100 C18 pre-column (3 μm, 100 Å, 75 μm × 2 cm, ThermoFisher Scientific, Reinach, Switzerland) and separated by backflush on a C18 column (3 μm, 100 Å, 75 μm × 15 cm, Nikkyo Technos, Tokyo, Japan) by applying a 60 min gradient of 5% to 40% acetonitrile in water and 0.1% formic acid, at a flow rate of 400 nL/min. Peptides of m/z 360–1400 were detected with resolution of 70,000, applying an automatic gain control (AGC) target of 10^6^ and a maximum ion injection time of 50 ms. A top ten data dependent method for precursor ion fragmentation with a stepped 27% normalized collision energy was applied with the following settings: precursor isolation width of 2 m/z, resolution 17,500, AGC of 10^5^ with a minimum target of 10^3^, maximum ion time of 110 ms, charge exclusion of unassigned and 1+ ions, peptide match on, and dynamic exclusion for 20 s, respectively. 

### 2.5. Measurement of RNA Expression Levels by Quantitative Reverse Transcriptase Real-Time PCR

To determine the RNA levels of selected *N. caninum* genes (see Table 1), quantitative reverse transcriptase RT-PCR was performed as described [12,22]. Briefly, 8 × 10^4^ HFF/well were seeded onto 6 well plates and cultured for 4 days at 37 °C/5% CO_2_ prior to infection with 6 × 10^5^
*N. caninum* tachyzoites/well. At 4 h post infection, the supernatant was aspirated and replaced with fresh medium containing either 5 µM BKI-1294 to form MNCs, or the corresponding amount of DMSO to generate tachyzoites. At 3 days post infection in the case of tachyzoites or 6 days post infection in the case of MNCs, RNA was isolated using the Nucleospin^®^ RNA Plus RNA isolation kit (Machery-Nagel, Düren, Germany) followed by cDNA synthesis using the Promega GoScript-Kit (Promega, Madison, WI, USA) according to the instructions provided by the manufacturers. Quantitative RT-PCR was performed using the Corbett Rotor gene 6000 (Qiagen, Venlo, Netherlands). The reactions contained 10 µL Sybr-Green-Mix, 0.1 µL of the forward and reverse primers listed in Table 1 and 10 µL of cDNA. 

### 2.6. Statistics

The MS data were obtained from three biological replicates, with two technical replicates for each biological replicate. All MS data were processed by MaxQuant (version 1.5.4.1) with matching between runs for the same strain activated, but not between different strains, in order to avoid over-interpretation of the data. The sample sets were interpreted separately by MaxQuant. Fragment spectra were interpreted against a recent *N. caninum* protein sequence database in fasta format (ToxoDB-44_NcaninumLIV_AnnotatedProteins.fasta). The trypsin cleavage rule allowed amide bond cleavage after lysine and arginine but not if a proline follows, up to three missed cleavage sites, fixed carbamidomethylation modification of cysteines, variable oxidation of methionine and acetylation of protein N-termini. Precursor and fragment mass tolerances were set to 10 and 20 ppm, respectively. Peptide spectrum matches and peptide and protein group identifications were filtered to a 1% false discovery rate (FDR) based on reversed database sequence matches, and a minimum of two razor or unique peptides were required to accept a protein group identification. Protein identifications considered as contaminations (e.g., trypsin or BSA) as well as proteins identified only by site (considered by MaxQuant developers as very likely false positives) were removed for statistical validation. The normalized label-free quantification (LFQ) protein group intensities as calculated by MaxQuant and the sum of the three most intense peptide intensities were used for relative proteome quantifications. First, we imputed missing protein LFQ values for samples in any condition group when there were at least two LFQ intensities in one group (downshift of 1.8 S.D. with a width of 0.3 S.D. [23]). Before summing, missing peptide intensities were imputed sample group wise, when there were at least 2 valid intensities (downshift of 1.8 S.D. with a width of 0.3 S.D.). The resulting protein intensities were named iTop3. This process left proteins without values in one or the other group. For Welch's T-tests, those missing protein intensities were replaced by imputed values from the very low end of intensity distributions in analogy to the above described imputation strategy (downshift 1.8 S.D., width of 0.3 S.D.). An FDR-controlled Benjamini-Hochberg procedure was used for correction of p-values. A log2-fold change of at least 1 and a corrected *p*-value of ≤0.05 were required to be considered as significant. Statistical testing and imputation were made with a homemade R (version 3.6.2 script run under R-Studio). 

## 3. Results

### 3.1. Formation of MNCs upon BKI-1294 Treatment In Vitro

Treatment of *N. caninum* infected HFF with 5 µM BKI-1294 lead to the formation of intracellular MNCs within 3–6 days, which are comparatively visualized with non-drug-treated tachyzoites in Figure 1. TEM of non-treated tachyzoites showed intracellular parasites undergoing endodyogeny, situated within a parasitophorous vacuole (Figure 1A). Figure 1C shows an MNC that formed after 6 days of treatment with BKI-1294. These MNCs were also located within a parasitophorous vacuole, were structurally intact, viable, and exhibited the typical features of apicomplexan parasites, but were not able to disconnect from each other and thus remained intracellular. SEM was performed on MNCs and non-treated tachyzoites that were separated from host cell debris by Sephadex G25 chromatography (Figure 1B,D). MNCs exhibited multiple newly formed apical complexes that emerged out of the MNC surface, as indicated in Figure 1D. 

### 3.2. The Proteome of Multinucleated Complexes Is Different from the Proteome of Tachyzoites

Shotgun mass spectrometry of the proteomes of *N. caninum* tachyzoites and of BKI-induced MNCs resulted in the identification of 207,041 unique peptides matching to 2558 proteins (Table 2). The complete dataset is given in Appendix A, which is available online.

By deeper analysis, we identified 1337 proteins that were differential between both stages with respect to both iTop3 and iLFQ parameters, as explained in Section 2.6. The large majority had lower iLFQ and iTop3 levels in MNCs as compared to tachyzoites. Only 12 proteins had higher levels (Table 2). 

Unbiased analysis of the dataset by principal component analysis (PCA) demonstrated for both iTop3 and iLFQ values (see Section 2.6) that the proteomes of tachyzoites and BKI-1294 induced MNCs formed non-overlapping clusters separated by principal component 1 (PC1). Moreover, the proteomes of biological and technical replicates clustered together for MNCs whereas the proteomes of biological and technical replicates of tachyzoites were separated along PC2 (Figure 2).

### 3.3. Multinucleated Complexes Express a Distinct Set of Potential Surface Antigens

A closer look at the 12 proteins with higher levels in MNCs compared to tachyzoites revealed that seven proteins had N-terminally located signal peptides, and six proteins had one or more SRS-domains (Table 3).

The relative abundance of these 12 proteins given by their respective iLFQ intensities—negligible in tachyzoites—reached up to 10^8^ iLFQ units in MNCs in the case of the SRS-domain containing protein NCLIV_019580, formerly annotated as SAG4 (Figure 3).

These values were, however, more than two orders of magnitude lower than the amount of the major surface antigen 1 of *N. caninum,* Nc SAG1 (NCLIV_033230). NcSAG1 reached 20.3 ± 1.3 × 10^9^ iLFQ units in tachyzoites and 19.9 ± 0.7 × 10^9^ iLFQ units in MNCs (Appendix A).

### 3.4. The SRS Domain Containing Protein NCLIV_019580 (NcSAG4) Has Higher mRNA Expression Levels in MNCs Compared to Tachyzoites

Since the SRS-domain containing protein encoded by NCLIV_019580, also annotated as NcSAG4, had the highest differential protein level in MNCs (see Figure 2), we investigated whether the NcSAG4 protein levels were correlated to the respective mRNA expression levels. We compared the mRNA levels in tachyzoites and MNCs of the corresponding coding sequence to N. caninum 28S RNA and included three other transcripts, namely mRNA coding for BAG1, IMC1, and CDPK1, the target of BKI-1294, into this comparison. As shown in Figure 4, MNCs had higher NcSAG4 mRNA levels compared to tachyzoites.

The same was found for the expression level of mRNA coding for NcBAG1. The absolute mRNA levels of BAG1 were, however, three magnitudes lower than those of the three other genes investigated. Conversely, the levels of mRNAs encoding CDPK1 and IMC1 were lower in MNCs than in tachyzoites.

## 4. Discussion

The proteome dataset generated for this study, comparing protein expression in BKI-1294-induced MNC and tachyzoites, contained 2558 non-redundant proteins, corresponding to 36% of the 7121 open reading frames identified in the *N. caninum* genome [24], thus nearly two times more than the number of proteins identified in a previous study [20].

In our comparative analysis of MNCs and tachyzoites, more than half of these proteins, including key enzymes of energy and intermediary metabolism (see Appendix A for details), are downregulated in MNCs induced by BKI-1294 treatment, and only twelve proteins are upregulated in MNCs. This suggests that MNCs may represent a resting state of the parasite that is induced by drug pressure exerted through BKI-1294 treatment.

This hypothesis is backed by the finding that mRNA coding for NcBAG1 (NCLIV_027470), also known as a heat-shock protein and a well-established marker for bradyzoites [25], can be detected in MNCs, but not in tachyzoites. The same has not only been observed in MNCs induced by BKI-1294 in a previous study [12], but also in intracellular parasites exposed to a different class of compounds, namely artemisinin derivatives such as artemisone and GC012 [22]. In addition, these findings were confirmed on the protein level by observing increased anti-BAG1 antibody staining in HFF infected with *N. caninum* and treated with these compounds [22]. The corresponding NcBAG1 protein is, however, not included in the proteome dataset (see Appendix A), suggesting that its expression levels are below the resolution limit of the method. Conversely, two other bradyzoite antigens, namely the SRS domain containing proteins NCLIV_010030, the bradyzoite antigen NcBSR4 [26], and NCLIV_019580, formerly annotated as NcSAG4, are amongst the twelve MNC-upregulated proteins. Of these, NcSAG4 exhibits the highest expression levels. 

The *N. caninum* SRS domain containing protein superfamily comprises 227 genes and 52 pseudogenes [24], considerably more than *Toxoplasma gondii* strains [27]. The most prominent of these proteins is certainly SAG1 (NCLIV_033230), the major surface antigen of *N. caninum*. During the last two decades, NcSAG1 has been investigated as a candidate for recombinant vaccine development [28,29] and for immunodiagnosis [30,31,32]. SAG1, considered an antigen expressed only in tachyzoites [33], is expressed at equally high levels in tachyzoites and MNCs, and therefore would not account for an immunological differentiation of both stages. Conversely, in our experiment, the bradyzoite-specific NcSAG4 [34] is expressed in MNCs only, most likely by regulation on the transcriptional level. Similarly, SAG4 expression has been shown to be upregulated in intracellular parasites treated with artemisinin derivatives such as artemisone and GC007 [22]. However, these derivatives have a different mode of action, and treatments of parasites induced completely different alterations. Nevertheless, also artemisone and GC007 required in vitro treatments at a concentration of 5 µM over a period of over 20 days in order to act parasiticidal, similar to what has been observed for BKI-1294 [12]. Thus, upregulation of NcSAG4 expression could be a general reaction to adverse physiological conditions, similar to what has been postulated for bradyzoites and tissue cyst formation [35].

Besides SAG4, six other MNC-induced proteins contain signal peptides and thus may constitute surface antigens, or secretory proteins involved in modulating the parasite-host interface. Five other proteins have SRS-domains. These overexpressed proteins may constitute suitable candidates for immunological investigations in vivo and—if this hypothesis could be confirmed—for recombinant vaccine development.

Constitutive expression of NcSAG4 in *N. caninum* tachyzoites had been shown to confer protection against vertical transmission in a mouse model [36]. Therefore, it is conceivable that the excellent protective effects of BKI-1294 and related compounds against vertical transmission [12,14,15,16] may be due to the induction of MNCs that express a set of diverse surface antigens, including NcSAG4. However, we currently do not know whether these antigens are expressed on the surface of newly formed zoites, or on the MNC surface. In any case, members of the SRS protein family are usually anchored to the plasma membrane by a glycosylphosphatidylinositol (GPI) molecule [37,38]. Besides NcSAG1 and NcSRS2 (also known as SRS29B and NcSRS29C, respectively), other SRS family members have been studied, including NcSRS67 that has no orthologue in *T. gondii* [39], and NcSRS57, which is expressed in both tachyzoites and bradyzoites [40,41]. However, according to our proteomics study, none of these appear to be overexpressed in MNCs.

Two additional studies described the development of similar multinucleated forms of the closely related *T. gondii*. Sugi et al. reported on another BKI named 1NM-PP1, which targeted *T. gondii* mitogen-activated protein kinase-like 1 (TgMAPKL-1), interfered in the cell cycle, and induced the formation enlarged multinucleated parasites. A mutation in a gatekeeper amino acid of TgMAPKL-1 led to a significant reduction in drug sensitivity and a reduction of enlarged multinucleated parasite formation [42]. In another study, a targeted deletion of the *rab11a* gene in *T. gondii* (TGME49_289680) also induced similar phenotypic changes [43]. The closest homologue in *N. caninum*, the unspecified protein NCLIV_041930, is expressed at similar levels in both tachyzoites and BKI-1294 induced MNCs (see Appendix A). 

Overall, we may consider that drug treatment, in the present case BKI-1294, does not only cause a shutdown of housekeeping gene expression and morphological alterations, but may also initiate “antigenic variation”, albeit to a much lesser extent than in other protozoan parasites such as the apicomplexan *Plasmodium* sp. [44], the Excavata *Trypanosoma* sp. [45,46] and *Giardia lamblia* [47]. MNC formation may thus constitute a response strategy to ensure the survival in the presence of BKIs and perhaps other compounds. Here, further investigations are required, also with respect to the infective potential of these MNCs. It is noteworthy to mention that similar multinucleated stages were induced upon in vitro treatment of *T. gondii* with diclazuril [48]. Diclazuril is triazinone derivative that is effective against intracellular stages of *Eimeria* and *Isospora spp*. While the mechanism of this drug is not well understood, it was shown that diclazuril is targeting enzymes of the respiratory chain and other enzymes such as dihydrofolate reductase (DHFR) [49]. However, we propose that MNCs represent a stage designated as baryzoites (Greek βαρυσ = massive, bulky, heavy, inert) as they represent a heavily enlarged parasitic stage that ensures survival of these parasites at increased drug concentrations during prolonged periods of time.

## Figures and Tables

**Figure 1 microorganisms-08-00801-f001:**
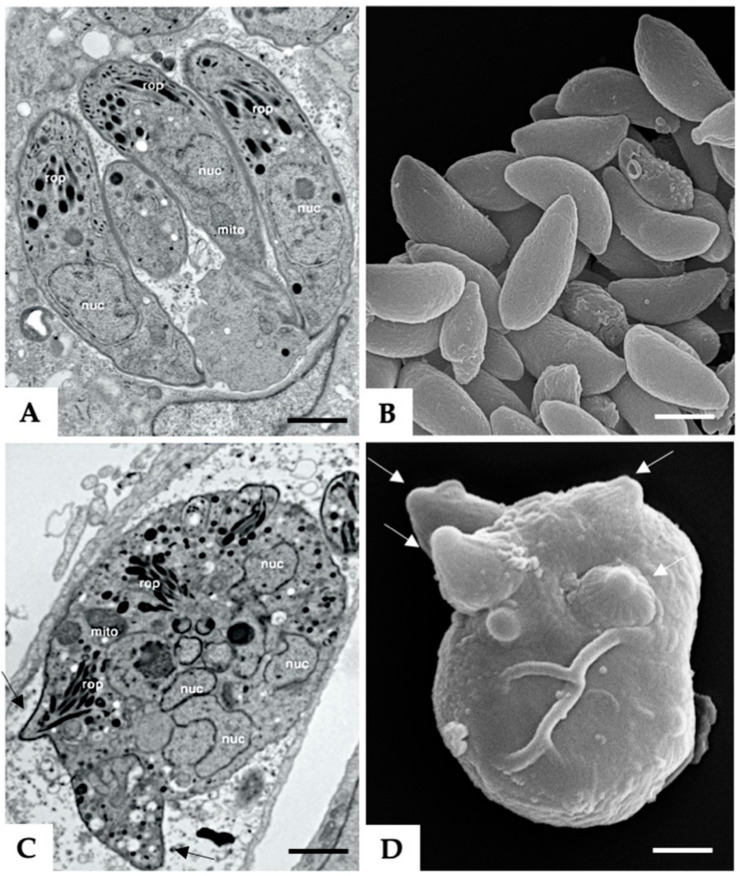
Tachyzoites (**A**,**B**) and BKI-1294 induced MNCs (**C**,**D**). (**A**,**C**) are TEM micrographs, with A showing individual tachyzoites situated within a parasitophorous vacuole, and © an MNC with multiple nuclei (nuc). Also visible are mitochondria (mito), rhoptries (rop), and apical complexes protruding from the MNC (black arrows). (**B**,**D**) are SEM micrographs of isolated tachyzoites and an MNC, respectively. Note the multiple apical complexes protruding from the surface of the MNC indicated by white arrows. Bar in (**A**) = 1 µm; bars in (**B**–**D**) = 1.2 µm.

**Figure 2 microorganisms-08-00801-f002:**
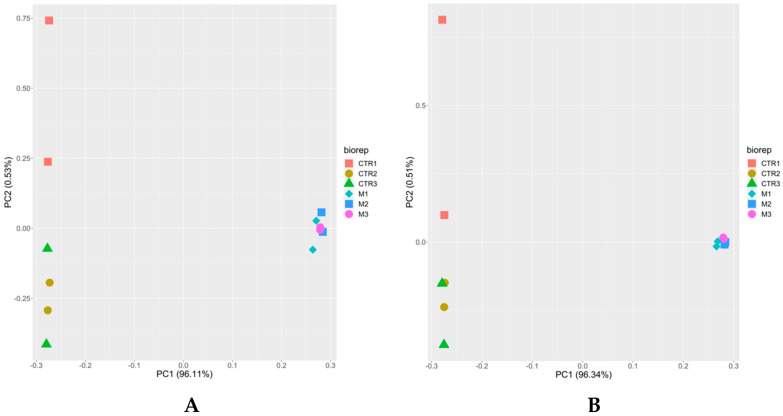
Principal component analysis of proteome data sets from *N. caninum* tachyzoites and from MNCs induced by BKI-1294 treatment. Tachyzoites and MNCs were grown, purified and subjected to MS shotgun analysis as described in the Materials and Methods. All technical and biological replicates are shown, namely red square, brown circle, and green triangle for tachyzoites (CTR), and pink circle, blue square, and turquoise diamond for MNCs (M). (**A**) iTop3 data; (**B**) iLFQ data.

**Figure 3 microorganisms-08-00801-f003:**
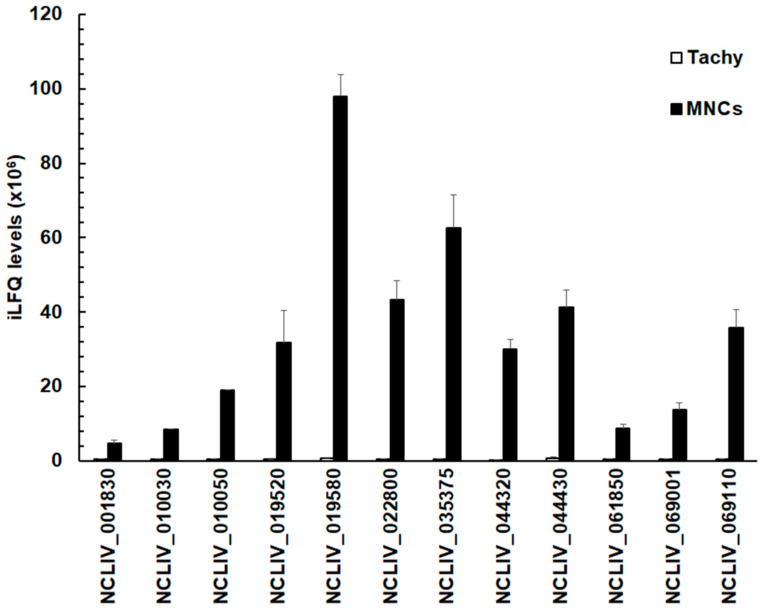
Quantitative assessments of the proteins in tachyzoites (Tachy, white bars) and MNCs (MNCs, black bars) listed in Table 3. For all proteins, mean values ± one standard deviation for LFQ intensities (×10^6^) in three biological replicates are shown. The proteins are termed by their respective accession numbers in the ToxoDB.

**Figure 4 microorganisms-08-00801-f004:**
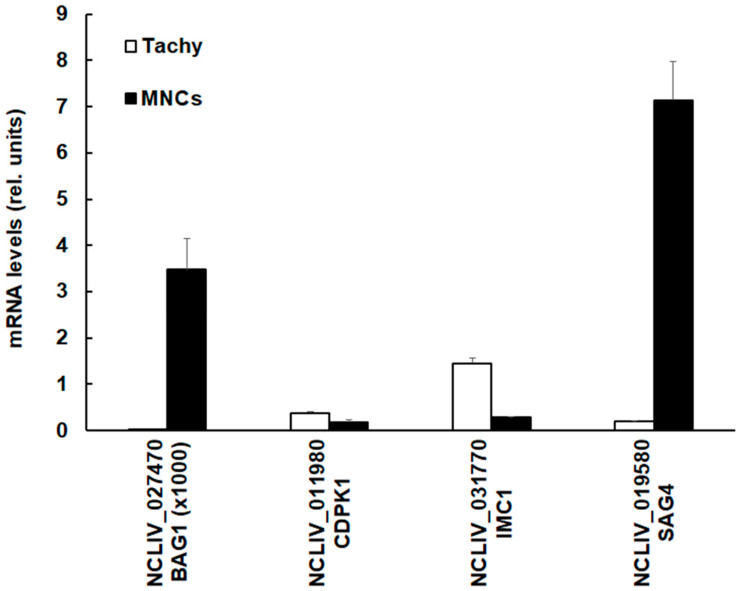
Quantitative assessments of mRNA levels of four selected genes. Confluent human foreskin fibroblast monolayers were infected with *N. caninum* tachyzoites and left untreated for the generation of tachyzoites (Tachy, white bars) or treated with BKI-1294 in order to generate MNCs (MNCs; black bars). Cells were harvested, mRNA was extracted, cDNA was prepared, and mRNA levels were quantified as described in Section 2.5. Mean ± standard error values from four replicates were assessed and expression levels were given as values in arbitrary units relative to the amount of 28S rRNA. The genes are termed by their respective accession numbers in ToxoDB.

**Table 1 microorganisms-08-00801-t001:** List of primers used in this study.

Annotation	Accession No	Primer	Sequence
Nc28SrRNA	L49389.1	Nc28S_F	TCTCTCTCACCAGGTTTAGG
		Nc28S_R	CCGTGTTTCAAGACGGGTC
Bradyzoite antigen, putative (BAG1)	NCLIV_027470	NcBAG1_F	CTCGACTTCATGGATGAGG
		NcBAG1_R	CTTCTATGGTAACGTCATCC
Calcium dependent protein kinase 1	NCLIV_011980	NcCDPKI_F	AGACGCTGCTATCGCGGG
		NcCDPKI_R	TTAGTTTCCGCAAAGCTTCAG
Inner membrane protein 1	NCLIV_031770	IMC1_F	GGATGGGACGGCGGTG
		IMC1_R	CGCCACCGGCTGGATG
SRS-domain containing protein	NCLIV_019580	Nc019580_F	CGCGTGCTCGGCGTGG
		Nc019580_R	GTGCTACCAGTTTGCATCC

**Table 2 microorganisms-08-00801-t002:** Summary of protein quantification data. *N. caninum* tachyzoites and MNCs were grown, purified and subjected to MS shotgun analysis as described in the Materials and Methods. For each group, three biological replicates have been tested (with 2 technical replicates per biological replicate). Only proteins with significant differences in both iTop3 and iLFQ levels were regarded as differential.

Parameter	Number
Unique peptides	207041
Non-redundant proteins	2558
Higher in tachyzoites	1325
Higher in MNCs	12
Similar levels in both	1221

**Table 3 microorganisms-08-00801-t003:** List of proteins overexpressed in MNCs formed by BKI-1294 treatment of *Neospora caninum* infected HFF monolayers. The NCLIV_ accession numbers in the ToxoDB database are given. IF, translation initiation factor; SP, signal peptide; SRS, SAG1-related surface antigen; TM, transmembrane.

Accession No	Annotation	Protein Features and Properties
001830	Hypothetical protein	SP; homologies to IF2 and collagen.
010030	Bradyzoite surface protein BSR4	SP; 2 SRS domains
010050	SRS domain-containing protein	2 SRS domains; homologies to 010030
019520	MGC83258 protein, related	1.7 MDa protein; homologies to ubiquitin ligase and mucin
019580	SRS domain-containing protein	SP, SRS domain, 2 TM domains, 1 lipoprotein domain, homology to *T. gondii*, bradyzoite surface antigen. Annotated as SAG4.
022800	Conserved hypothetical protein	Homologies to serine-rich adhesion and collagen
035375	SRS domain-containing protein	SP, 2 SRS domains
044320	Zinc finger protein 467, related	Homologies to serine-rich adhesins
044430	Hypothetical protein	Short sequences with homologies to zinc finger proteins and ATP-ase domain of Hsp90
061850	SRS domain-containing protein	SP, TM domain, 4 SRS domains.
069001	SRS domain-containing protein	SP, 2 SRS domains.
069110	Hypothetical protein	SP

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
