# Peer review of "Neospora caninum: Differential Proteome of Multinucleated Complexes Induced by the Bumped Kinase Inhibitor BKI-1294"

_microorganisms, 2020, doi:10.3390/microorganisms8060801_

Round 1
Reviewer 1 Report
In the manuscript, the authors compare the Neospora caninum multinucleate complexes (MMC) proteosome induced by a chemical compound (BKI-1294), with the evolutionary form of tachyzoite. The work is very interesting, it is well written, however some comments/ suggestions / clarifications need to be presented:
- Would the BKI-1294 compound really be a pharmacological candidate for the development of anti-Neospora drugs? This compound induces the formation of MMC, a "third evolutionary form", called by the authors "Baryzoites". Would this third evolutionary form of the parasite be viable? If so, would it not be another escape mechanism for the parasite to induce parasitism in the host?
- Is the "Baryzoites" evolutionary form, when isolated, capable of inducing infection in a susceptible animal host? I think it could be an important issue for the discussion of the manuscript.
- Page 2 (lines 65-67): It would be important for the authors to cite references that state that kinase genes (CDPKs) originate from plants.
- Page 2 (line 84): "BKI-1294... but is not parasiticidal" - In this context, how do the authors believe that this compound can be an important and rational antiparasitic drug?
- The authors propose a third evolutionary form for Neospora caninum. What would be the characteristics of this third evolutionary form and the role in the parasite's biological cycle? I think it would be equally important to raise these issues in the discussion.
Author Response
Responses to reviewer 1:
Thank you for your constructive comments on our manuscript.
In the manuscript, the authors compare the Neospora caninum multinucleate complexes (MMC) proteosome induced by a chemical compound (BKI-1294), with the evolutionary form of tachyzoite. The work is very interesting, it is well written, however some comments/ suggestions / clarifications need to be presented:
- Would the BKI-1294 compound really be a pharmacological candidate for the development of anti-Neospora drugs? This compound induces the formation of MMC, a "third evolutionary form", called by the authors "Baryzoites". Would this third evolutionary form of the parasite be viable? If so, would it not be another escape mechanism for the parasite to induce parasitism in the host?
Our response: BKI-1294 is indeed a candidate for the development of anti-Neospora drugs, and as it does not interfere in pregnancy neither in the mouse model nor in pregnant sheep infected with the closely related Toxoplasma gondii. This is outlined in the introduction, lanes 79-82: This compound has shown highly promising efficacy in pregnant mouse models for neosporosis [12] and for toxoplasmosis [13], and in a pregnant sheep model for toxoplasmosis [14]. Moreover, other BKIs are effective against neosporosis, as shown in pregnant mouse [15] and sheep [16] models.
- Is the "Baryzoites" evolutionary form, when isolated, capable of inducing infection in a susceptible animal host? I think it could be an important issue for the discussion of the manuscript.
Our response: we have not tested this explicitly to date, but the question is very valid and should be investigated in the future. In order to survive, these parasites have to invade a host cell. Host cell invasion has been studied most extensively for the tachyzoite stage, and is a complex multistep process that involves three secretory organelles, namely rhoptries, micronemes and dense granules. Since these baryzoites are rather large, it is unlikely for them to achieve host cell entry easily. In contrast, we suggest that these baryzoites represent a resting stage that allows the parasite to survive despite constant drug pressure (see Introduction, lanes 87-88). These results suggested that MNCs could constitute a drug-induced resting stage that assures the survival of the parasite until the drug pressure is released. We also mention this in the discussion, lanes 383-385: MNC formation may thus constitute a response strategy to ensure the survival in the presence of BKIs and perhaps other compounds. Here, further investigations are required, also with respect to the infective potential of these MNCs.
- Page 2 (lines 65-67): It would be important for the authors to cite references that state that kinase genes (CDPKs) originate from plants.
Our response: this was done (lane 64)
- Page 2 (line 84): "BKI-1294... but is not parasiticidal" - In this context, how do the authors believe that this compound can be an important and rational antiparasitic drug?
Our response: On one hand, we have cited a number of studies in the preceding paragraph (lanes 74-77) which have earlier demonstrated excellent efficacy of BKI-1294 in vivo. This compound has shown highly promising efficacy in pregnant mouse models for neosporosis [12] and for toxoplasmosis [13], and in a pregnant sheep model for toxoplasmosis [14]. Moreover, other BKIs are effective against neosporosis, as shown in pregnant mouse [15] and sheep [16] models. We also changed the sentence on page 2, lane 78 ("BKI-1294... but is not parasiticidal") to BKI-1294... but does not act parasiticidal in vitro to distinguish from in vivo results. We believe, that MNC formation could impact on immunity, which would then contribute to compound efficacy, but this is pure speculation and we will have to investigate this hypothesis in the future
- The authors propose a third evolutionary form for Neospora caninum. What would be the characteristics of this third evolutionary form and the role in the parasite's biological cycle? I think it would be equally important to raise these issues in the discussion.
Our response: We do not think that baryzoites play a role in the normal biological cycle of this or related parasites. As indicated above, baryzoite formation occurs upon drug pressure, in this case BKI-1294. However, similar multinucleated complexes have been described in earlier studies, but have never been investigated in such detail before. An example is the action of diclazuril against T. gondii. We added a corresponding remark on this fact at the end of the discussion (lanes 386-390): It is noteworthy to mention that similar multinucleated stages were induced upon in vitro treatment of T. gondii with diclazuril [48]. Diclazuril is triazinone derivative that is effective against intracellular stages of Eimeria and Isospora spp. While the mechanism of this drug is not well understood, it was shown that diclazuril is targeting enzymes of the respiratory chain and other enzymes such as dihydrofolate reductase (DHFR) [49].
Reviewer 2 Report
I have carefully examined the manuscript entitled “Neospora caninum: Differential proteome of multinucleated complexes induced by the bumped kinase inhibitor BKI-1294” by Winzer et al.
In this work, authors apply the shotgun proteomic approach to measure the proteome expression of Neospora caninum, a parasite causing problems in farm animals, in its different stages. In particular, they compare the protein different expression between tachyzoite and bradyzoite stages with that of the Bumped Kinase inhibitor BKI-1294-induced intracellular multinucleated complexes (MNCs), and demonstrate MNCs as a novel viable stage of the parasite.
Authors took advantage of their solid experience and knowledge on apicomplexan parasites, as resulted by their previous studies, to perform an accurate and well-done proteomic analysis. They compared protein levels, and in particular the overexpressed proteins, between tachyzoite, bradyzoite and MNCs stages with the purpose to identify their specific biomarkers.
Authors performed shotgun proteomics and statistics, detecting a consistent number of proteins differentially expressed between tachyzoite and MNCs, and among them only 12 up-regulated in MNCs, mainly connected to surface antigenic properties.
The results of the proteomic approach were solid and of high quality, and support the proposal of MNCs as a new stage of N. caninum, due to survival response to drug pressure.
On this basis, I consider the results interesting and also significant in the field of apicomplexan parasite field. Moreover, the manuscript doesn’t need language revision.
I have only a criticism, regarding the publication in the past few days of the article “Neospora caninum: Structure and Fate of Multinucleated Complexes Induced by the Bumped Kinase Inhibitor BKI-1294” on Pathogens by (mostly) the same authors. Indeed, after reading this paper I found that most of the introduction part is almost superimposable with that of the present manuscript, and I suggest the authors to re-write this part of the manuscript accordingly, also adding in the introduction part a reference of the Pathogens paper.
For these reasons I believe that the above-mentioned manuscript will be suitable for publication on Microorganisms once the authors will revise the manuscript answering to my comment.
Author Response
Responses to reviewer 2:
Thank you for your constructive comments on our manuscript.
We have no rewritten a large portion of the Introduction as requested. Changes are marked in red font.
In addition, we have added the reference Winzer et al., 2020 (Pathogens 9, 382). At the time this paper was submitted, that paper was not accepted yet.